# Sequence-to-Sequence Learning with Latent Neural Grammars

**Yoon Kim**
MIT CSAIL
`yoonkim@mit.edu`

## Abstract

Sequence-to-sequence learning with neural networks has become the de facto standard for sequence prediction tasks. This approach typically models the local distribution over the next word with a powerful neural network that can condition on arbitrary context. While flexible and performant, these models often require large datasets for training and can fail spectacularly on benchmarks designed to test for compositional generalization. This work explores an alternative, hierarchical approach to sequence-to-sequence learning with quasi-synchronous grammars, where each node in the target tree is transduced by a node in the source tree. Both the source and target trees are treated as latent and induced during training. We develop a neural parameterization of the grammar which enables parameter sharing over the combinatorial space of derivation rules without the need for manual feature engineering. We apply this latent neural grammar to various domains—a diagnostic language navigation task designed to test for compositional generalization (SCAN), style transfer, and small-scale machine translation—and find that it performs respectably compared to standard baselines.

## 1   Introduction

Sequence-to-sequence learning with neural networks [62, 22, 106] encompasses a powerful and general class of methods for modeling the distribution over an output target sequence $y$ given an input source sequence $x$. Key to its success is a factorization of the output distribution via the chain rule coupled with a richly-parameterized neural network that models the local conditional distribution over the next word given the previous words and the input. While architectural innovations such as attention [8], convolutional layers [39], and Transformers [110] have led to significant improvements, this word-by-word modeling remains core to the approach, and with good reason—since any distribution over the output can be factorized autoregressively via the chain rule, this approach should be able to well-approximate the true target distribution given large-enough data and model.[1]

However, despite their excellent performance across key benchmarks these models are often sample inefficient and can moreover fail spectacularly on diagnostic tasks designed to test for compositional generalization [68, 63]. This is partially attributable to the fact that standard sequence-to-sequence models have relatively weak inductive biases (e.g. for capturing hierarchical structure [79]), which can result in learners that over-rely on surface-level (as opposed to structural) correlations.

In this work, we explore an alternative, hierarchical approach to sequence-to-sequence learning with *latent neural grammars*. This work departs from previous approaches in three ways. First, we model the distribution over the target sequence with a *quasi-synchronous grammar* [103] which assumes a hierarchical generative process whereby each node in the target tree is transduced by

---

Much of the work was completed while the author was at MIT-IBM Watson AI. Code is available at `https://github.com/yoonkim/neural-qcfg`.

[1]There are, however, weighted languages whose next-word conditional distributions are hard to compute in a formal sense, and these distributions cannot be captured by locally normalized autogressive models unless one allows the number of parameters (or runtime) to grow superpolynomially in sequence length [72].

35th Conference on Neural Information Processing Systems (NeurIPS 2021).

nodes in the source tree. Such node-level alignments provide provenance and a causal mechanism for how each output part is generated, thereby making the generation process more interpretable. We additionally find that the explicit modeling of source- and target-side hierarchy improves compositional generalization compared to non-hierarchical models. Second, in contrast the existing line of work on incorporating (often observed) tree structures into sequence modeling with neural networks [35, 5, 89, 37, 126, 1, 97, 18, 34, *inter alia*], we treat the source and target trees as fully *latent* and induce them during training. Finally, whereas previous work on synchronous grammars typically utilized log-linear models over handcrafted/pipelined features [20, 56, 115, 103, 112, 27, 42, *inter alia*] we make use of *neural* features to parameterize the grammar's rule probabilities, which enables efficient sharing of parameters over the combinatorial space of derivation rules without the need for any manual feature engineering. We also use the grammar directly for end-to-end generation instead of as part of a larger pipelined system (e.g. to extract alignments) [122, 41, 14].

We apply our approach to a variety of sequence-to-sequence learning tasks—SCAN language navigation task designed to test for compositional generalization [68], style transfer on the English Penn Treebank [78], and small-scale English-French machine translation—and find that it performs respectably compared to baseline approaches.

## 2 Neural Synchronous Grammars for Sequence-to-Sequence Learning

We use $\boldsymbol{x} = x_1, \ldots, x_S$, $\boldsymbol{y} = y_1, \ldots, y_T$ to denote the source/target strings, and further use $\boldsymbol{s}, \boldsymbol{t}$ to refer to source/target trees, represented as a set of nodes including the leaves (i.e. $\mathrm{yield}(\boldsymbol{s}) = \boldsymbol{x}$ and $\mathrm{yield}(\boldsymbol{t}) = \boldsymbol{y}$).

### 2.1 Quasi-Synchronous Grammars

Quasi-synchronous grammars, introduced by Smith and Eisner [103], define a monolingual grammar over target strings conditioned on a source tree, where the grammar's rule set depends dynamically on the source tree $\boldsymbol{s}$. In this paper we work with probabilistic quasi-synchronous context-free grammars (QCFG), which can be represented as a tuple $G[\boldsymbol{s}] = (S, \mathcal{N}, \mathcal{P}, \Sigma, \mathcal{R}[\boldsymbol{s}], \theta)$ where $S$ is the distinguished start symbol, $\mathcal{N}$ is the set of nonterminals which expand to other nonterminals, $\mathcal{P}$ is the set of nonterminals which expand to terminals (i.e. preterminals), $\Sigma$ is the set of terminals, and $\mathcal{R}[\boldsymbol{s}]$ is a set of context-free rules conditioned on $\boldsymbol{s}$, where each rule is one of

$$
\begin{aligned}
&S \rightarrow A[\alpha_i], && A \in \mathcal{N}, \quad \alpha_i \subseteq \boldsymbol{s} \\
&A[\alpha_i] \rightarrow B[\alpha_j]C[\alpha_k], && A \in \mathcal{N}, \quad B, C \in \mathcal{N} \cup \mathcal{P}, \quad \alpha_i, \alpha_j, \alpha_k \subseteq \boldsymbol{s} \\
&D[\alpha_i] \rightarrow w, && D \in \mathcal{P}, w \in \Sigma, \quad \alpha_i \subseteq \boldsymbol{s}.
\end{aligned}
$$

We use $\theta$ to parameterize the rule probabilities $p_\theta(r)$ for each $r \in \mathcal{R}[\boldsymbol{s}]$. In the above, $\alpha_i$'s are subsets of nodes in the source tree $\boldsymbol{s}$, and thus a QCFG tranduces the output tree by aligning each target tree node to a subset of source tree nodes. This monolingual generation process differs from that of classic synchronous context-free grammars [118] which jointly generate source and target trees in tandem (and therefore require that source and target trees be isomorphic), making QCFGs appropriate tools for tasks such as machine translation where syntactic divergences are common.[2] Since the $\alpha_i$'s are elements of the power set of $\boldsymbol{s}$, the above formulation as presented is completely intractable. We follow prior work [103, 112] and restrict $\alpha_i$'s to be single nodes (i.e. $\alpha_i, \alpha_j, \alpha_k \in \boldsymbol{s}$), which amounts to assuming that each target tree node is aligned to exactly one source tree node.

In contrast to standard, "flat" sequence-to-sequence models where any hierarchical structure necessary for the task must be captured implicitly within a neural network's hidden layers, synchronous grammars explicitly model the hierarchical structure on both the source and target side, which acts as a strong source of inductive bias. This tree transduction process furthermore results in a more interpretable generation process as each span in the target aligned to a span in the source via node-level alignments.[3] More generally, the grammar's rules provide a symbolic interface to the model with which operationalize constraints and imbue inductive biases, and we show how this mechanism can be used to, for example, incorporate phrase-level copy mechanisms (section 2.4).

---

[2]It is also possible to model syntactic divergences with richer grammatical formalisms [101, 81]. However these approaches require more expensive algorithms for learning and inference.

[3]Similarly, latent variable attention [121, 7, 28, 96, 119] provides for more a interpretable generation process than standard soft attention via explicit word-level alignments.

## 2.2 Parameterization

Since each source tree node $\alpha_i$ is likely to occur only a few times (or just once) in the training corpus, parameter sharing becomes crucial. Prior work on QCFGs typically utilized log-linear models over handcrafted features to share parameters across rules [103, 42]. In this work we instead use a neural parameterization which allows for easy parameter sharing without the need for manual feature engineering. Concretely, we represent each target nonterminal and source node combination $A[\alpha_i]$ as an embedding,

$$\mathbf{e}_{A[\alpha_i]} = \mathbf{u}_A + \mathbf{h}_{\alpha_i},$$

where $\mathbf{u}_A$ is the embedding for $A$, and $\mathbf{h}_{\alpha_i}$ is the representation of node $\alpha_i$ given by running a TreeLSTM over the source tree $s$ [107, 134]. These embeddings are then combined to produce the probability of each rule,

$$
\begin{aligned}
p_\theta(S \to A[\alpha_i]) &\propto \exp\left(\mathbf{u}_S^\top \mathbf{e}_{A[\alpha_i]}\right), \\
p_\theta(A[\alpha_i] \to B[\alpha_j]C[\alpha_k]) &\propto \exp\left(f_1(\mathbf{e}_{A[\alpha_i]})^\top (f_2(\mathbf{e}_{B[\alpha_j]}) + f_3(\mathbf{e}_{C[\alpha_k]}))\right), \\
p_\theta(D[\alpha_i] \to w) &\propto \exp\left(f_4(\mathbf{e}_{D[\alpha_i]})^\top \mathbf{u}_w + b_w\right),
\end{aligned}
$$

where $f_1, f_2, f_3, f_4$ are feedforward networks with residual layers (see Appendix A.1 for the exact parameterization). Therefore the learnable parameters in this model are the nonterminal embeddings (i.e. $\mathbf{u}_A$ for $A \in \{S\} \cup \mathcal{N} \cup \mathcal{P}$), terminal embeddings/biases (i.e. $\mathbf{u}_w, b_w$ for $w \in \Sigma$), and the parameters of the TreeLSTM and the feedforward networks.

## 2.3 Learning and Inference

The QCFG described above defines a distribution over target trees (and by marginalization, target strings) given a source tree. While prior work on QCFGs typically relied on an off-the-shelf parser over the source to obtain its parse tree, this limits the generality of the approach. In this work, we learn a probabilistic source-side parser along with the QCFG. This parser is a monolingual PCFG with parameters $\phi$ that defines a posterior distribution over binary parse trees given source strings, i.e. $p_\phi(s \,|\, \boldsymbol{x})$. Our PCFG uses the neural parameterization from Kim et al. [64]. With the parser in hand, we are now ready to define the log marginal likelihood,

$$\log p_{\theta,\phi}(\boldsymbol{y} \,|\, \boldsymbol{x}) = \log\left(\sum_{s \in \mathcal{T}(\boldsymbol{x})} \sum_{t \in \mathcal{T}(\boldsymbol{y})} p_\theta(t \,|\, s) p_\phi(s \,|\, \boldsymbol{x})\right).$$

Here $\mathcal{T}(\boldsymbol{x})$ and $\mathcal{T}(\boldsymbol{y})$ are the sets of trees whose yields are $\boldsymbol{x}$ and $\boldsymbol{y}$ respectively. Unlike in synchronous context-free grammars, it is not possible to efficiently marginalize over both $\mathcal{T}(\boldsymbol{y})$ and $\mathcal{T}(\boldsymbol{x})$ due to the non-isomorphic assumption. However, we observe that the inner summation $\sum_{t \in \mathcal{T}(\boldsymbol{y})} p_\theta(t \,|\, s) = p_\theta(\boldsymbol{y} \,|\, s)$ can be computed with the usual inside algorithm [9] in $\mathcal{O}(|\mathcal{N}|(|\mathcal{N}| + |\mathcal{P}|)^2 S^3 T^3)$, where $S$ is the source length and $T$ is the target length. This motivates the following lower bound on the log marginal likelihood,

$$\log p_{\theta,\phi}(\boldsymbol{y} \,|\, \boldsymbol{x}) \geq \mathbb{E}_{s \sim p_\phi(s \,|\, \boldsymbol{x})}\left[\log p_\theta(\boldsymbol{y} \,|\, s)\right],$$

which is obtained by the usual application of Jensen's inequality (see Appendix A.2).[4]

An unbiased Monte Carlo estimator for the gradient with respect to $\theta$ is straightforward to compute given a sample from $p_\phi(s \,|\, \boldsymbol{x})$, since we can just backpropagate through the inside algorithm. For the gradient respect to $\phi$, we use the score function estimator with a self-critical baseline [92],

$$\nabla_\phi \mathbb{E}_{s \sim p_\phi(s \,|\, \boldsymbol{x})}\left[\log p_\theta(\boldsymbol{y} \,|\, s)\right] \approx (\log p_\theta(\boldsymbol{y} \,|\, s') - \log p_\theta(\boldsymbol{y} \,|\, \widehat{s})) \nabla_\phi \log p_\phi(s' \,|\, \boldsymbol{x}),$$

---

[4]As is standard in variational approaches, one can tighten this bound with the use of a variational distribution $q_\psi(s \,|\, \boldsymbol{x}, \boldsymbol{y})$, which results in the following evidence lower bound,

$$\log p_{\theta,\phi}(\boldsymbol{y} \,|\, \boldsymbol{x}) \geq \mathbb{E}_{s \sim q_\psi(s \,|\, \boldsymbol{x}, \boldsymbol{y})}\left[\log p_\theta(\boldsymbol{y} \,|\, s)\right] - \mathrm{KL}[q_\psi(s \,|\, \boldsymbol{x}, \boldsymbol{y}) \,\|\, p_\phi(s \,|\, \boldsymbol{x})].$$

This is equivalent to our objective if we set $q_\psi(s \,|\, \boldsymbol{x}, \boldsymbol{y}) = p_\phi(s \,|\, \mathbf{x})$. Rearranging some terms, we then have,

$$\mathbb{E}_{s \sim p_\phi(s \,|\, \boldsymbol{x})}\left[\log p_\theta(\boldsymbol{y} \,|\, s)\right] = \log p_{\theta,\phi}(\boldsymbol{y} \,|\, \boldsymbol{x}) - \mathrm{KL}[p_\phi(s \,|\, \boldsymbol{x}) \,\|\, p_{\theta,\phi}(s \,|\, \boldsymbol{x}, \boldsymbol{y})].$$

Hence, our use of $p_\phi(s \,|\, \mathbf{x})$ as the variational distribution is militating towards learning a model which achieves good likelihood but at the same time has a posterior distribution $p_{\theta,\phi}(s \,|\, \boldsymbol{x}, \boldsymbol{y})$ that is close to the prior $p_\phi(s \,|\, \mathbf{x})$ (i.e. learning a model where most of the uncertainty about $s$ is captured by $\boldsymbol{x}$ alone). This is arguably reasonable for many language applications since parse trees are often assumed to be task-agnostic.

where $\boldsymbol{s}'$ is a sample from $p_\phi(\boldsymbol{s} \mid \boldsymbol{x})$ and $\widehat{\boldsymbol{s}}$ is the MAP tree from $p_\phi(\boldsymbol{s} \mid \boldsymbol{x})$. We also found it important to regularize the source parser by simultaneously training it as a monolingual PCFG, and therefore add $\nabla_\phi \log p_\phi(\boldsymbol{x})$ to the gradient expression above.[5] Obtaining the sample tree $\boldsymbol{s}'$, the argmax tree $\widehat{\boldsymbol{s}}$, and scoring the sampled tree $\log p_\phi(\boldsymbol{s}' \mid \boldsymbol{x})$ all require $\mathcal{O}(S^3)$ dynamic programs. Hence the runtime is still dominated by the $\mathcal{O}(S^3 T^3)$ dynamic program to compute $\log p_\theta(\boldsymbol{y} \mid \boldsymbol{s}')$ and $\log p_\theta(\boldsymbol{y} \mid \widehat{\boldsymbol{s}})$.[6] We found this to be manageable on modern GPUs with a vectorized implementation of the inside algorithm. Our implementation uses the Torch-Struct library [93].

**Predictive inference** For decoding, we first run MAP inference with the source parser to obtain $\widehat{\boldsymbol{s}} = \operatorname{argmax}_{\boldsymbol{s}} p_\phi(\boldsymbol{s} \mid \boldsymbol{x})$. Given $\widehat{\boldsymbol{s}}$, finding the most probable sequence $\operatorname{argmax}_{\boldsymbol{y}} p_\theta(\boldsymbol{y} \mid \widehat{\boldsymbol{s}})$ (i.e. the consensus string of the grammar $G[\widehat{\boldsymbol{s}}]$) is still difficult, and in fact NP-hard [102, 16, 77]. We therefore resort to an approximate decoding scheme where we sample $K$ target trees $\boldsymbol{t}^{(1)}, \ldots \boldsymbol{t}^{(K)}$ from $G[\widehat{\boldsymbol{s}}]$, rescore the yields of the sampled trees, and return the tree whose yield has the lowest perplexity.

## 2.4 Extensions

Here we show that the formalism of synchronous grammars provides a flexible interface with which to interact with the model.

**Phrase-level copying** Incorporating copy mechanisms into sequence-to-sequence models has led to significant improvements for tasks where there is overlap between the source and target sequences [58, 83, 48, 73, 47, 95]. These models typically define a latent variable at each time step that learns to decide to either copy from the source or generate from the target vocabulary. While useful, most existing copy mechanisms can only copy singletons due to the word-level encoder/decoder.[7] In contrast, the hierarchical generative process of QCFGs makes it convenient to incorporate phrase-level copy mechanisms by using a special-purpose nonterminal/preterminal that always copies the yield of the source subtree that it is combined with. Concretely, letting $A_{\mathrm{COPY}} \in \mathcal{N}$ be a COPY nonterminal, we can expand the rule set $\mathcal{R}[\boldsymbol{s}]$ to include rules of the form $A_{\mathrm{COPY}}[\alpha_i] \to v$ for $v \in \Sigma^+$, and define the probabilities to be,

$$p_\theta(A_{\mathrm{COPY}}[\alpha_i] \to v) \stackrel{\mathrm{def}}{=} \mathbb{1}\{v = \mathrm{yield}(\alpha_i)\}.$$

(The preterminal copy mechanism is similarly defined.) Computing $p_\theta(\boldsymbol{y} \mid \boldsymbol{s})$ in this modified grammar requires a straightforward modification of the inside algorithm.[8] In our style transfer experiments in section 3.2 we show that this phrase-level copying is important for obtaining good performance. While not explored in the present work, such a mechanism can readily be employed to incorporate external transformations rules (e.g. from bilingual lexicons or transliteration tables) into the modeling process, which has been previously investigated at the singleton-level [88, 3].

**Adding constraints on rules** For some applications we may want to place additional restrictions on the rule set to operationalize domain-specific constraints and inductive biases.  For example, setting $\alpha_j, \alpha_k \in \mathrm{descendant}(\alpha_i)$ for rules of the form $A[\alpha_i] \to B[\alpha_j]C[\alpha_k]$ would constrain the target tree hierarchy to respect the source tree hierarchy, while restricting $\alpha_i$ to source terminals (i.e. $\alpha_i \in \mathrm{yield}(\boldsymbol{s})$) for rules of the form $D[\alpha_i] \to w$ would enforce that each target terminal be aligned to a source terminal. We indeed make use of such restrictions in our experiments.

**Incorporating autoregressive language models** Finally, we remark that simple extensions of the QCFG can incorporate standard autoregressive language models. Let $p_{\mathrm{LM}}(w \mid \gamma)$ be a distribution over the next word given by a (potentially conditional) language model given arbitrary context $\gamma$ (e.g. $\gamma = \boldsymbol{y}_{<t}$ for a monolingual language model and $\gamma = \boldsymbol{x}, \boldsymbol{y}_{<t}$ for a sequence-to-sequence model). One way to embed this language model into a QCFG would be to use a special LM preterminal $D_{\mathrm{LM}} \in \mathcal{P}$ that is not combined with any source node, and define the emission probability to be,

$$p_\theta(D_{\mathrm{LM}} \to w) \stackrel{\mathrm{def}}{=} p_{\mathrm{LM}}(w \mid \gamma).$$

(The nonterminal probabilities $p_\theta(A[\alpha_i] \to D_{\mathrm{LM}}C[\alpha_k])$ and $p_\theta(A[\alpha_i] \to B[\alpha_j]D_{\mathrm{LM}})$ are computed with the associated symbol embedding $\mathbf{u}_{D_{\mathrm{LM}}}$.) Both the QCFG and the language model can then be trained jointly.

---

[5]This motivates our use of a generative rather than a discriminative parser on the source side.

[6]This runtime is incidentally is the same as that of the bitext inside algorithm for marginalizing over both source and target trees in rank-two synchronous context-free grammars.

[7]However see Zhou et al. [132], Panthaplackel et al. [87], and Wiseman et al. [114].

[8]Letting $\beta[s, t, N] = p_\theta(N \stackrel{*}{\to} \boldsymbol{y}_{s:t})$ be the inside variable for $N$'s being the root of the subtree over $\boldsymbol{y}_{s:t}$, we can simply set $\beta[s, t, A_{\mathrm{COPY}}[\alpha_i]] = \mathbb{1}\{\boldsymbol{y}_{s:t} = \mathrm{yield}(\alpha_i)\}$.

| Approach | Simple | Jump | A. Right | Length |
|---|---|---|---|---|
| RNN [68] | 99.7 | 1.7 | 2.5 | 13.8 |
| CNN [29] | 100.0 | 69.2 | 56.7 | 0.0 |
| Transformer [38] | – | 1.0 | 53.3 | 0.0 |
| T5-base [38] | – | 99.5 | 33.2 | 14.4 |
| Syntactic Attn [94] | 100.0 | 91.0 | 28.9 | 15.2 |
| Meta Seq2Seq [67] | – | 99.9 | 99.9 | 16.6 |
| CGPS [70] | 99.9 | 98.8 | 83.2 | 20.3 |
| Equivar. Seq2Seq [44] | 100.0 | 99.1 | 92.0 | 15.9 |
| Span-based SP [54] | 100.0 | – | 100.0 | – |
| LANE [76] | 100.0 | 100.0 | 100.0 | 100.0 |
| Program Synth. [85] | 100.0 | 100.0 | 100.0 | 100.0 |
| NeSS [19] | 100.0 | 100.0 | 100.0 | 100.0 |
| NQG-T5 [98] | 100.0 | 100.0 | – | 100.0 |
| GECA [6] | – | 87.0 | 82.0 | – |
| R&R Data Aug. [4] | – | 88.0 | 82.0 | – |
| Neural QCFG (ours) | 96.9 | 96.8 | 98.7 | 95.7 |

**Table 1:** Accuracy on the SCAN dataset splits compared to previous work.

$P_0[\text{run}] \rightarrow \text{RUN}$
$P_0[\text{look}] \rightarrow \text{LOOK}$
$P_0[\text{walk}] \rightarrow \text{WALK}$
$P_0[\text{jump}] \rightarrow \text{JUMP}$
$P_0[\text{right}] \rightarrow \text{TURN-RIGHT}$
$P_0[\text{left}] \rightarrow \text{TURN-LEFT}$
$N_4[\text{look left}] \rightarrow P_0[\text{left}]\ P_0[\text{look}]$
$N_4[\text{look right}] \rightarrow P_0[\text{right}]\ P_0[\text{look}]$
$N_4[\text{walk left}] \rightarrow P_0[\text{left}]\ P_0[\text{walk}]$
$N_4[\text{walk right}] \rightarrow P_0[\text{right}]\ P_0[\text{walk}]$
$N_1[\text{look right twice}] \rightarrow N_4[\text{look right}]\ N_4[\text{look right}]$
$N_1[\text{walk left twice}] \rightarrow N_4[\text{walk left}]\ N_4[\text{walk left}]$
$N_1[\text{look thrice}] \rightarrow N_8[\text{look thrice}]\ P_0[\text{look}]$
$N_1[\text{look right thrice}] \rightarrow N_8[\text{look right thrice}]\ N_4[\text{look right}]$
$N_8[\text{look right thrice}] \rightarrow N_4[\text{look right}]\ N_4[\text{look right}]$
$N_1[\text{walk left thrice}] \rightarrow N_8[\text{walk left thrice}]\ N_4[\text{walk left}]$
$N_8[\text{walk left thrice}] \rightarrow N_4[\text{walk left}]\ N_4[\text{walk left}]$

**Table 2:** Frequently-occurring rules from MAP target trees on the *add primitive (jump)* train set.

For some cases we may want to make use of a conditional language model that condition on subparts of the source sentence. This may be appropriate for learning to translate non-compositional phrases whose translations cannot be obtained by stitching together independent translations of subparts (e.g. idioms such as "kicked the bucket"). In this case we can make use of a special nonterminal $A_{\text{LM}} \in \mathcal{N}$ which is combined with source tree nodes to produce rules of the form $A_{\text{LM}}[\alpha_i] \rightarrow v$ for $v \in \Sigma^+$. The associated probabilities are then defined to be,

$$p_\theta(A_{\text{LM}}[\alpha_i] \rightarrow v) \stackrel{\text{def}}{=} p_{\text{LM}}(v \mid \text{yield}(\alpha_i)).$$

While these extensions can embed flexible autoregressive models within a QCFG,[9] they also inherit many of the issues attendant with such models (e.g. over-reliance on surface form). In preliminary experiments with these variants, we found the combined model to quickly degenerate into the uninteresting case of always using the conditional language model, and hence did not pursue this further. However it is possible that modifications to the approach (e.g. posterior regularization to penalize overuse of the conditional language model) could lead to improvements.

## 3  Experiments

We apply the neural QCFG described above to a variety of sequence-to-sequence learning tasks. These experiments are not intended to push the state-of-the-art on these tasks but rather intended to assess whether our approach performs respectably against standard baselines while simulatenously learning interesting and interpretable structures.

### 3.1  SCAN

We first experiment on SCAN [68], a diagnostic dataset where a model has to learn to translate simple English commands to actions (e.g. jump twice after walk $\rightarrow$ WALK JUMP JUMP). While conceptually simple, standard sequence-to-sequence models have been shown to fail on splits of the data designed to test for compositional generalization. We focus on four commonly-used splits: (1) *simple*, where train/test split is random, (2) *add primitive (jump)*, where the primitive command jump is seen in isolation in training and must combine with other commands during testing,[10] (3) *add template (around right)*, where the template around right is not seen during training, and (4) *length*, where the model is trained on action sequences of length at most 22 and tested on action sequences of length between 24 and 48.

---

[9]Alternatively, we can also embed a QCFG within an autoregressive language model with a binary latent variable $z_t$ (with distribution $p_{\text{LM}}(z_t \mid \boldsymbol{x}, \boldsymbol{y}_{<t})$) at each time step. This variable—marginalized over during training—selects between $p_{\text{LM}}(y_t \mid \boldsymbol{x}, \boldsymbol{y}_{<t})$ and $p_\theta(y_t \mid \boldsymbol{s}, \boldsymbol{y}_{<t})$, where the latter next-word probability distribution in the QCFG can be computed with a probabilistic Earley parser [105]. The difference between the approaches stems from whether the switch decision is made by the QCFG or by the language model.

[10]The QCFG defined in this paper places zero probability on length-one target strings, which presents an issue for this split of SCAN where jump $\rightarrow$ JUMP is the only context in which "JUMP" occurs in the training set. To address this, in cases where the target string is a singleton we simply replicate the source and target, i.e. jump $\rightarrow$ JUMP becomes jump jump $\rightarrow$ JUMP JUMP.

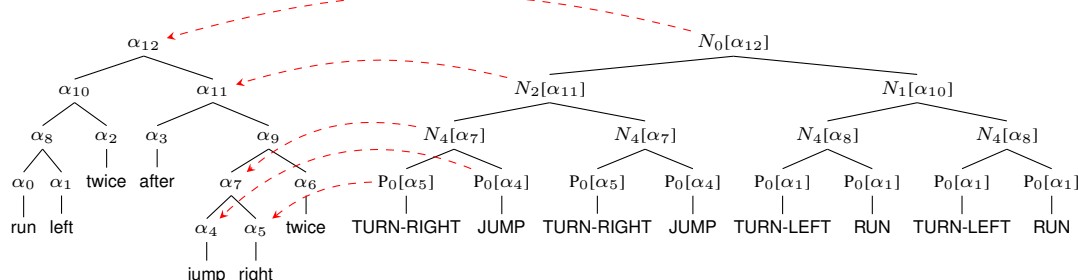

**Figure 1:** Generation from the neural QCFG on a test example from the *add primitive (jump)* split of SCAN. The induced tree from the learned source parser is shown on the left, and the target tree derivation is shown on the right. We do not show the initial root-level node (i.e. $S \rightarrow N_0[\alpha_{12}]$). While the model does not distinguish between preterminals and terminals on the source tree, we have shown them separately for additional clarity. We also show some of the node-level alignments with dashed lines.

In these experiments, the nonterminals $A \in \mathcal{N}$ are only combined with source nodes that govern at least two nodes, and the preterminals $P \in \mathcal{P}$ are only combined with source terminals. We set $|\mathcal{N}| = 10$ and $|\mathcal{P}| = 1$, and place two additional restrictions on the rule set. First, for rules of the form $S \rightarrow A[\alpha_i]$ we restrict $\alpha_i$ to always be the root of the source tree. Second, for rules of the form $A[\alpha_i] \rightarrow B[\alpha_j]C[\alpha_k]$ we restrict $\alpha_j, \alpha_k$ either be a descendant of $\alpha_i$, or $\alpha_i$ itself (i.e. $\alpha_j, \alpha_k \in \text{descendant}(\alpha_i) \cup \{\alpha_i\}$). These restrictions operationalize the constraint that the target tree hierarchy respects the source tree hierarchy, though still in a much looser sense than in an isomorphism. We found these constraints to be crucial in learning models that perform well on the compositional splits of the dataset. See Appendix A.3.1 for the full experimental setup and hyperparameters.

**Results**  Table 1 shows our results against various baselines on SCAN. While many approaches are able to solve this dataset almost perfectly, they often make use of SCAN-specific knowledge, which precludes their straightforward application to non-synthetic domains. The neural QCFG performs respectably while remaining domain-agnostic. In Table 2 we show some examples of frequently-occurring rules based on their MAP target tree counts on the training set of the *add primitive (jump)* split. Many of the rules are sensible, and they furthermore illustrate the need for multiple nonterminal symbols. For example, in order to deal with source phrases of form "*x* thrice" in a grammar that only has unary and binary rules, the model uses the nonterminals $N_1$ and $N_8$ in different ways when combined with the same phrase. Figure 1 shows an example generation from the test set of the *add primitive (jump)* split, where we find that node-level alignments provide explicit provenance for each target span and thus makes the generation process more interpretable than standard attention mechanisms. These alignments can also be used to diagnose and rectify systematic errors. For example, we sometimes found the model to incorrectly split "*x* {and,after} *y*" to "*x x*" (or "*y y*") at the root node. When we manually disallowed such splits during decoding, performance increased by 1%-2% across the board, showcasing a benefit of grammar-based models which makes it possible to directly manipulate model generations by intervening on the set of derivation rules.

### 3.2 Style Transfer

We next apply our approach on style transfer on English utilizing the StylePTB dataset from Lyu et al. [78]. We focus on the three *hard transfer* tasks identified by the original paper: (1) *active to passive*, where a sentence has to be changed from active to passive voice (2808 examples), (2) *adjective emphasis*, where a sentence has to be rewritten to emphasize a particular adjective (696 examples) (3) *verb/action emphasis*, where a sentence has to be rewritten to emphasize a particular verb/action (1201 examples).[11] The main difficulty with these tasks stems from the small training set combined with the relative complexity of these tasks.

For these experiments we set $|\mathcal{N}| = |\mathcal{P}| = 8$ and use the same restrictions on the rule set as in the SCAN experiments. We also found it helpful to contextualize the source embedding with a bidirectional LSTM before feeding to the TreeLSTM encoder.[12] We further experiment with the

---

[11]To encode information about which word to be emphasized in the adjective/verb emphasis tasks, we use a binary variable whose embedding is added to the word embedding on the encoder side.

[12]A drawback of using contextualized word embeddings as input to the TreeLSTM is that since the representations $\mathbf{h}_{\alpha_i}$ for each node $\alpha_i$ are now a function of the entire sentence (and not just the leaves), we can no

| Transfer Type | Approach | BLEU-1 | BLEU-2 | BLEU-3 | BLEU-4 | METEOR | ROUGE-L | CIDEr |
|---|---|---|---|---|---|---|---|---|
| Active to Passive | GPT2-finetune | 0.476 | 0.329 | 0.238 | 0.189 | 0.216 | 0.464 | 1.820 |
| | Seq2Seq | 0.373 | 0.220 | 0.141 | 0.103 | 0.131 | 0.345 | 0.845 |
| | Retrieve-Edit | 0.681 | 0.598 | 0.503 | 0.427 | 0.383 | 0.663 | 4.535 |
| | Human | 0.931 | 0.881 | 0.835 | 0.795 | 0.587 | 0.905 | 8.603 |
| | Seq2Seq | 0.505 | 0.349 | 0.253 | 0.190 | 0.235 | 0.475 | 2.000 |
| | Neural QCFG | 0.431 | 0.637 | 0.548 | 0.472 | 0.415 | 0.695 | 4.294 |
| | Seq2Seq + copy | 0.838 | 0.735 | 0.673 | 0.598 | 0.467 | 0.771 | 5.941 |
| | Neural QCFG + copy | 0.836 | 0.771 | 0.713 | 0.662 | 0.499 | 0.803 | 6.410 |
| Adj. Emphasis | GPT2-finetune | 0.263 | 0.079 | 0.028 | 0.000 | 0.112 | 0.188 | 0.386 |
| | Seq2Seq | 0.187 | 0.058 | 0.018 | 0.000 | 0.059 | 0.179 | 0.141 |
| | Retrieve-Edit | 0.387 | 0.276 | 0.211 | 0.164 | 0.193 | 0.369 | 1.679 |
| | Human | 0.834 | 0.753 | 0.679 | 0.661 | 0.522 | 0.811 | 6.796 |
| | Seq2Seq | 0.332 | 0.333 | 0.051 | 0.000 | 0.142 | 0.27 | 0.845 |
| | Neural QCFG | 0.348 | 0.178 | 0.062 | 0.000 | 0.162 | 0.317 | 0.667 |
| | Seq2Seq + copy | 0.505 | 0.296 | 0.184 | 0.119 | 0.242 | 0.514 | 1.839 |
| | Neural QCFG + copy | 0.676 | 0.506 | 0.393 | 0.316 | 0.373 | 0.683 | 3.424 |
| Verb Emphasis | GPT2-finetune | 0.309 | 0.170 | 0.095 | 0.041 | 0.140 | 0.292 | 0.593 |
| | Seq2Seq | 0.289 | 0.127 | 0.066 | 0.038 | 0.098 | 0.275 | 0.300 |
| | Retrieve-Edit | 0.416 | 0.284 | 0.209 | 0.148 | 0.223 | 0.423 | 1.778 |
| | Human | 0.649 | 0.569 | 0.493 | 0.421 | 0.433 | 0.693 | 5.668 |
| | Seq2Seq | 0.355 | 0.152 | 0.083 | 0.043 | 0.151 | 0.320 | 0.530 |
| | Neural QCFG | 0.431 | 0.250 | 0.140 | 0.073 | 0.219 | 0.408 | 1.097 |
| | Seq2Seq + copy | 0.526 | 0.389 | 0.294 | 0.214 | 0.294 | 0.464 | 2.346 |
| | Neural QCFG + copy | 0.664 | 0.512 | 0.407 | 0.319 | 0.370 | 0.589 | 3.227 |

**Table 3:** Results on the *hard* style transfer tasks from the StylePTB dataset [78]. For each transfer type, the top four rows are from Lyu et al. [78], while the bottom four rows are from this paper. Metrics such as BLEU and ROUGE are normally scaled to [0, 100] (as in Table 4), but here we keep them at [0, 1] as in the original paper.

phrase-level copy mechanism as described in section 2.4. The original paper provides several strong baselines: finetuned GPT2, a standard sequence-to-sequence model, and the retrieve-and-edit model from Hashimoto et al. [53]. We also train our own baseline sequence-to-sequence model with a word-level copy mechanism. See Appendix A.3.2 for more details.

**Results** Table 3 shows the results where we observe that the neural QCFG performs well compared to the various baselines.[13] We further find that incorporating the copy mechanism improves results substantially for both the baseline LSTM and the neural QCFG.[14] Figure 2 shows a test example from the *active-to-passive* task, which shows the word- and phrase-level copying mechanism in action. In this example the source tree is linguistically incorrect, but the grammar is nonetheless able to appropriately transduce the output. Given that linguistic phrases are generally more likely to remain unchanged in these types tasks, incorporating this knowledge into the learning process could potentially improve results.[15] For example in Figure 2 the ideal case would be to copy the phrase "a 2-for-1 stock split" but this is not possible due to the incorrectly predicted source tree. Finally, although our approach ostensibly improves upon the baselines according to many of the $n$-gram-based metrics, we observed the generated sentences to be often ungrammatical, highlighting the limitations of automatic metrics for these tasks while at the same time indicating opportunities for further work in this area.

---

longer guarantee that target derivations such as $A[\alpha_i] \xrightarrow{*} \boldsymbol{y}_{s:t}$ only depend on $\alpha_i$. This somewhat hinders the interpretability of the node-level alignments.

[13]As in the original paper we calculate the automatic metrics using the nlg-eval library, available at https://github.com/Maluuba/nlg-eval.

[14]Even for models that do not explicitly use the copy mechanism, we indirectly allow for copying by replacing the ⟨unk⟩ token with the source token that the preterminal is combined with in the neural QCFG case, or the source token that had the maximum attention weight in the LSTM case. This explains the outperformance of our baseline sequence-to-sequence models compared to the baselines from Lyu et al. [78], which roughly uses the same architecture.

[15]There are many ways to do this. For example, one could identify the longest overlap between the source and target, and use posterior regularization on the source PCFG to encourage it to be a valid constituent.

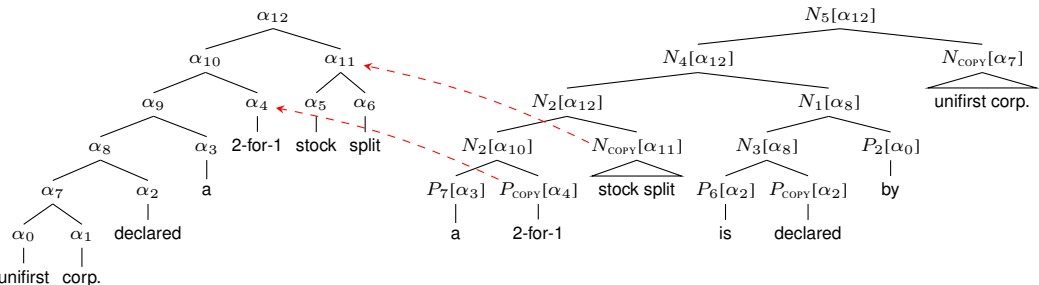

**Figure 2:** A test example from the *active to passive* style transfer task on the Penn Treebank. The induced tree from the learned source parser is shown on the left, and the target tree derivation is shown on the right. The source tree is linguistically incorrect but the model is still able to correctly tranduce the output. Some examples of COPY nonterminals/preterminals and their aligned source nodes are shown with dashed arrows.

### 3.3 Machine Translation

Our final experiment is on a small-scale English-French machine translation dataset from Lake and Baroni [68]. Here we are interested in evaluating the model in two ways: first, to see if it can perform well as a standard machine translation system on a randomly held out test set, and second, to see if it can systematically generalize to unseen combinations. To assess the latter, Lake and Baroni [68] add 1000 repetitions of i am daxy → je suis daxiste to the training set and test on 8 new sentences that use *daxy* in novel combinations (e.g. he is daxy → il est daxiste and i am not daxy → je ne suis pas daxiste). As the original dataset does not provide official splits, we randomly split the dataset into 6073 examples for training (1000 of which is the "i am daxy" example), 631 examples for validation, and 583 for test.[16]

For these experiments, we set $|\mathcal{N}| = |\mathcal{P}| = 14$ and combine all source tree nodes with all nonterminals/preterminals. We place two restrictions on the rule set: for rules of the form $S \to A[\alpha_i]$ we restrict $\alpha_i$ to be the root of the source tree (as in previous experiments), and for rules of the form $A[\alpha_i] \to B[\alpha_j]C[\alpha_k]$ we restrict $\alpha_j, \alpha_k$ to be the direct children of $\alpha_i$ such that $\alpha_j \neq \alpha_k$ (or $\alpha_i$ itself if $\alpha_i$ has no children).[17] As in the style transfer experiments, we also experiment with a bidirectional LSTM encoder which contextualizes the source word embeddings before the TreeLSTM layer. Our baselines here include standard LSTM/Transformer models as well as approaches that explicitly target compositional generalization [70, 19].

| Approach | BLEU | *daxy* acc. |
|---|---|---|
| LSTM | 25.1 | 12.5% |
| Transformer | 30.4 | 100% |
| CGPS [70] | 19.2 | 100% |
| NeSS [19] | – | 100% |
| Neural QCFG | 23.5 | 100% |
| + BiLSTM | 26.8 | 75.0% |

**Table 4:** Results on English-French machine translation.

**Results** Table 4 shows BLEU on the regular test set of 583 sentences and accuracy on the 8 *daxy* sentences.[18] While the neural QCFG performs nontrivially, it is soundly outperformed by a well-tuned Transformer model, which performs impressively well even on the *daxy* test set.[19] We thus consider our results on machine translation to be largely negative. Interestingly, the use of contextualized word embeddings (via the bidirectional LSTM) improves BLEU but hurts compositional generalization, highlighting the potential pitfalls of using flexible models which can sometimes entangle representations in undesirable ways.[20] Figure 3 shows several examples of target tree derivations from the neural QCFG that does

---

[16]The original dataset has 10000 examples (not including the *daxy* examples), but many of them involve duplicate source sentences. We removed such duplicates in our split of the data.

[17]These restrictions are closer to the strict isormorphic requirement in synchronous context-free grammars than in previous experiments. However they still allow for non-isormorphic trees since $\alpha_i$ can be inherited if it has no children.

[18]For CGPS and NeSS, the original papers only assess accuracy on the *daxy* test set, and furthermore do not provide the training/validation/test splits. To obtain BLEU for CGPS, we run the publicly available code (https://github.com/yli1/CGPS) on our split of the data. For NeSS, the code is not publicly available but the authors provided a version of their implementation. However, the provided code/hyperparameters were tailored for the SCAN dataset, and despite our best efforts to adapt the code/hyperparameters to our setup we were unable obtain sensible results on the machine translation dataset.

[19]The Transformer did, however, require some hyperparameter tuning given the small size of our dataset. Similar findings have been reported by Wu et al. [120] in the context of applying Transformers to moderately sized character-level transduction datasets.

[20]This variant of the neural QCFG also does poorly on the compositional splits of SCAN.

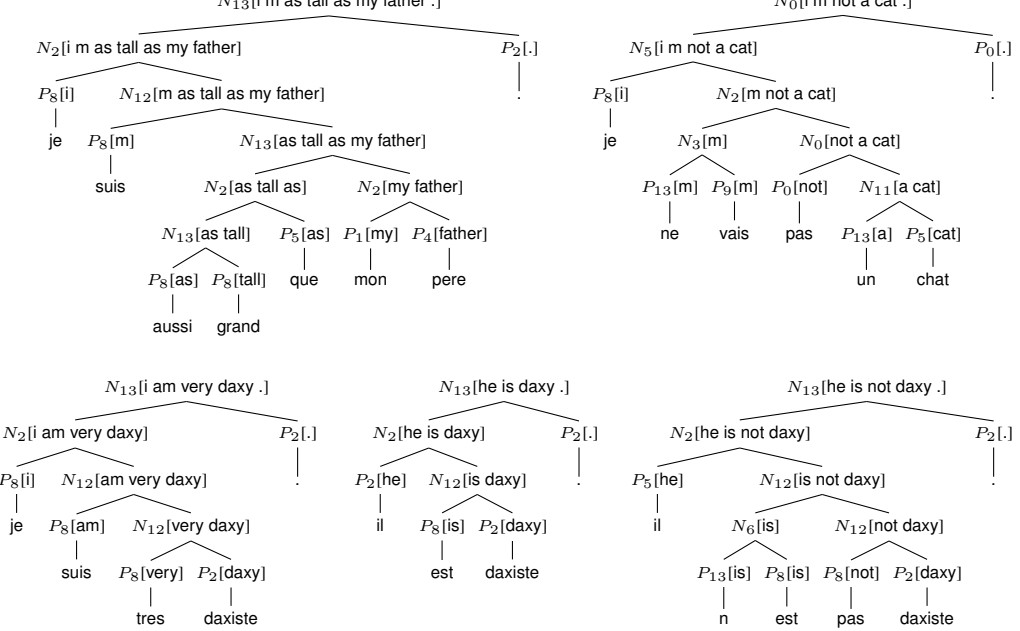

**Figure 3:** Target tree derivations from the English-French machine translation experiments. Top left is an example from the regular test set, top right is a made-up example (which is incorrectly translated by the model), and the bottom three trees are from the *daxy* test set. We do not explicitly show the source trees here and instead show the source phrases as arguments to the target tree nonterminals/preterminals.

not use contextualized word embeddings. The induced source trees are sometimes linguistically incorrect (e.g. in the top left example "as tall as" would not be considered a valid linguistic phrase), but the QCFG is still able to correctly transduce the output by also learning unconventional trees on the target side as well. This is reminiscent of classic hierarchical phrase-based approaches to machine translation where the extracted phrases often do not correspond to linguistic phrases [20]. Finally, although the model does well on the *daxy* test set, it still incorrectly translates simple but unusual made-up examples such as "i m not a cat" (Figure 3, top right). This is despite the fact that examples of the form i {am,m} not a $x \rightarrow$ je ne suis pas {un,une} $y$ occur multiple times in the training set.[21] We speculate that while the probabilistic nature of the grammar and the use of distributed representations enables easier training, they contribute to the model's being (still) vulnerable to spurious correlations.

## 4 Discussion

While we have shown that neural quasi-synchronous grammars can perform well for some sequence-to-sequence learning tasks, there are several serious limitations. For one, the $\mathcal{O}(|\mathcal{N}|(|\mathcal{N}| + |\mathcal{P}|)^2 S^3 T^3)$ dynamic program will likely pose challenges in scaling this approach to larger datasets with longer sequences.[22] Predictive inference was also much more expensive since we found it necessary to sample and score a large number of target trees to perform well on the non-synthetic datasets (see Appendix A.3). The conditional independence assumptions made by the QCFG may also be too strong for some tasks that involve complex dependencies, and the approach may furthermore be inappropriate for domains where the input and output are not naturally tree-structured. The models were quite sensitive to hyperparameters and some datasets needed training over multiple random seeds to perform well. These factors make our approach much less "off-the-shelf" than standard sequence-to-sequence models, although this may be partially attributable to the availability of a robust set of hyperparameters for existing approaches.

---

[21]The other models were also unable to correctly translate this sentence.

[22]Indeed, on realistic machine translation datasets with longer sequences we quickly ran into memory issues when running the model on just a single example, even with a multi-GPU implementation of the inside algorithm distributed over four 32GB GPUs. To apply the model on longer sentences, an interesting future direction might involve working with a "soft" version of the grammar, where the nonterminals embeddings are contexualized against source elements via soft attention. The runtime and memory for marginalizing over target trees in this soft QCFG would have a linear (instead of cubic) dependence source length.

At the start of this project, our initial hope was to show that classic, grammar-based approaches to sequence transduction had been unfairly overlooked in the current deep learning era, and that revisiting these methods with contemporary parameterizations would prove to be more than just an academic exercise. Disappointingly, this seems not to be the case. While we did observe decent performance on niche datasets such as SCAN and StylePTB where inductive biases from grammars were favorably aligned to the task at hand, for tasks like machine translation our approach was thoroughly steamrolled by a well-tuned Transformer.

What role, then, can such models play in building practical NLP systems (if any)? It remains to be seen, but we venture some guesses. Insofar as grammars and other models with symbolic components are able to better surface model decisions than standard approaches, they may have a role in developing more controllable and interpretable models, particularly in the context of collaborative human-machine systems [40]. Alternatively, inflexible models with strong inductive biases have in the past been used to guide (overly) flexible neural models in various ways, for example by helping to generate additional data [58, 75] or inducing structures with which to regularize/augment models [24, 74, 3, 127]. In this vein, it may be interesting to explore how induced structures from grammars (such as the tranduction rules in Table 2) can be used in conjunction with flexible neural models.

## 5 Related Work

**Synchronous grammars** Synchronous grammars and tree transducers have a long and rich history in natural language processing [2, 101, 118, 80, 36, 32, 84, 56, 115, 45, 12, 23, *inter alia*]. In this work we focus on the formalism of quasi-synchronous grammars, which relaxes the requirement that source trees be isomorphic to target trees. Quasi-synchronous grammars have enjoyed applications across a wide range of domains including in machine translation [103, 42, 43], question answering [112], paraphrase detection [27], sentence simplification [117, 116], and parser projection [104]. Prior work on quasi-synchronous grammars generally relied on pipelined parse trees for the source and only marginalized out the target tree, in contrast to the present work which treats both source and target trees as latent.

**Compositional sequence-to-sequence learning** Lake and Baroni [68] proposed the influential SCAN dataset for assessing the compositional generalization capabilities of neural sequence-to-sequence models. There has since been a large body of work on compositional sequence-to-sequence learning through various approaches including modifications to existing architectures [70, 94, 44, 17, 26], grammars and neuro-symbolic models [86, 98, 85, 19, 76], meta-learning [67, 25], and data augmentation [6, 49, 50, 4]. Our approach is closely related to NQG-T5 [98] which uses a rules-based approach to induce a non-probabilistic QCFG and then backs off to a flexible sequence-to-sequence model during prediction if the grammar cannot parse the input sequence.

**Deep latent variable models** There has much work on neural parameterizations of classic probabilistic latent variable models including hidden Markov models, [109, 113, 21], finite state transducers [91, 71] topic models [82, 30, 31], dependency models [59, 51, 15, 52, 123], and context-free grammars [64, 61, 133, 131, 125, 124]. These works essentially extend feature-based unsupervised learning [10] to the neural case with the use of neural networks over embedding parameterizations, which makes it easy to share parameters and additionally condition the generative model on side information such as auxiliary latent variables [52, 64], images [130, 60, 55], video and audio [129], and source-side context [113, 99]. Since we marginalize over unobserved trees during learning, our work is also related to the line of work on marginalizing out latent variables/structures for sequence transduction tasks [46, 33, 11, 66, 128, 90, 57, 69, 108, 111, *inter alia*].

## 6 Conclusion

In this paper we have studied sequence-to-sequence learning with latent neural grammars. We have shown that the formalism quasi-synchronous grammars provides a flexible tool with which to imbue inductive biases, operationalize constraints, and interface with the model. Future work in this area could consider: (1) revisiting richer grammatical formalisms (e.g. synchronous tree-adjoining grammars [101]) with contemporary parameterizations, (2) conditioning on other modalities such as images/audio for grounded grammar induction [100, 130, 60, 129], (3) adapting these methods to other structured domains such as programs and graphs, and (4) investigating how grammars and symbolic models can be integrated with pretrained language models to solve practical tasks.

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
