# A Appendix

## A.1 Neural QCFG Parameterization

Each nonterminal is combined with a source node to produce a symbol $A[\alpha_i]$, whose embedding representation is given by $\mathbf{e}_{A[\alpha_i]} = \mathbf{u}_A + \mathbf{h}_{\alpha_i}$. Here $\mathbf{u}_A$ is a randomly initialized embedding and $\mathbf{h}_{\alpha_i}$ is the node representation for $\alpha_i$ from a TreeLSTM. The rule probabilities are then given by,

$$p_\theta(S \to A[\alpha_i]) = \frac{\exp\left(\mathbf{u}_S^\top \mathbf{e}_{A[\alpha_i]}\right)}{\sum\limits_{A' \in \mathcal{N}} \sum\limits_{\alpha' \in s} \exp\left(\mathbf{u}_S^\top \mathbf{e}_{A'[\alpha']}\right)},$$

$$p_\theta(A[\alpha_i] \to B[\alpha_j]C[\alpha_k]) = \frac{\exp\left(f_1(\mathbf{e}_{A[\alpha_i]})^\top (f_2(\mathbf{e}_{B[\alpha_j]}) + f_3(\mathbf{e}_{C[\alpha_k]}))\right)}{\sum\limits_{B' \in \mathcal{M}} \sum\limits_{\alpha' \in s} \sum\limits_{C'' \in \mathcal{M}} \sum\limits_{\alpha'' \in s} \exp\left(f_1(\mathbf{e}_{A[\alpha_i]})^\top (f_2(\mathbf{e}_{B'[\alpha']}) + f_3(\mathbf{e}_{C''[\alpha'']}))\right)},$$

$$p_\theta(D[\alpha_i] \to w) = \frac{\exp\left(f_4(\mathbf{e}_{D[\alpha_i]})^\top \mathbf{u}_w + b_w\right)}{\sum\limits_{w' \in \Sigma} \exp\left(f_4(\mathbf{e}_{D[\alpha_i]})^\top \mathbf{u}_{w'} + b_{w'}\right)},$$

where $\mathcal{M} = \mathcal{N} \cup \mathcal{P}$. In the above $f_i$'s are feedforward networks with three residual blocks,

$$f_i(\mathbf{x}) = g_{i,3}(g_{i,2}(g_{i,1}(\mathbf{W}_i\mathbf{x} + \mathbf{b}_i))), \qquad i \in \{1, 2, 3, 4\},$$
$$g_{i,j}(\mathbf{x}) = \text{ReLU}(\mathbf{V}_{i,j}(\text{ReLU}(\mathbf{U}_{i,j} + \mathbf{c}_{i,j})) + \mathbf{d}_{i,j}) + \mathbf{x}, \qquad j \in \{1, 2, 3\}.$$

We often place restrictions on the derivations to operationalize domain-specific constraints. For example, in our machine translation experiments we constrain $\alpha_j, \alpha_k$ to be the immediate children of $\alpha_j$ for rules of the form $A[\alpha_i] \to B[\alpha_j]C[\alpha_k]$ such that $\alpha_j \neq \alpha_k$, unless $\alpha_i$ is a leaf node in which case $\alpha_i$ is always inherited. To calculate $p_\theta(A[\alpha_i] \to B[\alpha_j]C[\alpha_k])$ with this restriction, we consider different cases. In the case where $\alpha_j, \alpha_k$ are the immediate children of $\alpha_i$, $p_\theta(A[\alpha_i] \to B[\alpha_j]C[\alpha_k])$ is given by

$$\frac{\exp\left(f_1(\mathbf{e}_{A[\alpha_i]})^\top (f_2(\mathbf{e}_{B[\alpha_j]}) + f_3(\mathbf{e}_{C[\alpha_k]}))\right)}{\sum\limits_{B' \in \mathcal{M}} \sum\limits_{C'' \in \mathcal{M}} \exp\left(f_1(\mathbf{e}_{A[\alpha_i]})^\top (f_2(\mathbf{e}_{B'[\alpha_j]}) + f_3(\mathbf{e}_{C''[\alpha_k]}))\right) + \exp\left(f_1(\mathbf{e}_{A[\alpha_i]})^\top (f_2(\mathbf{e}_{B'[\alpha_k]}) + f_3(\mathbf{e}_{C''[\alpha_j]}))\right)}.$$

In the case where $\alpha_i \in \text{yield}(s)$ (i.e. it has no children), we have

$$p_\theta(A[\alpha_i] \to B[\alpha_j]C[\alpha_k]) = \begin{cases} \frac{\exp\left(f_1(\mathbf{e}_{A[\alpha_i]})^\top (f_2(\mathbf{e}_{B[\alpha_j]}) + f_3(\mathbf{e}_{C[\alpha_k]}))\right)}{\sum\limits_{B' \in \mathcal{M}} \sum\limits_{C'' \in \mathcal{M}} \exp\left(f_1(\mathbf{e}_{A[\alpha_i]})^\top (f_2(\mathbf{e}_{B'[\alpha_i]}) + f_3(\mathbf{e}_{C''[\alpha_i]}))\right)}, & \alpha_j = \alpha_k = \alpha_i, \\ 0, & \text{otherwise.} \end{cases}$$

All other $p_\theta(A[\alpha_i] \to B[\alpha_j]C[\alpha_k])$'s are assigned to 0. In practice these constraints are implemented by masking out the full third-order tensor of the rules' (log) probabilities, which is of size $(2S - 1)|\mathcal{N}| \times (2S - 1)|\mathcal{N} \cup \mathcal{P}| \times (2S - 1)|\mathcal{N} \cup \mathcal{P}|$. Here $(2S - 1)$ is the number of nodes in the binary source tree. This masking strategy makes it possible to still utilize vectorized implementations of the inside algorithm.

## A.2 Lower Bound Derivation

$$\begin{aligned} \log p_{\theta,\phi}(\boldsymbol{y} \mid \boldsymbol{x}) &= \log \left( \sum_{\boldsymbol{s} \in \mathcal{T}(\boldsymbol{x})} \sum_{\boldsymbol{t} \in \mathcal{T}(\boldsymbol{y})} p_\theta(\boldsymbol{t} \mid s) p_\phi(\boldsymbol{s} \mid \boldsymbol{x}) \right) \\ &= \log \left( \sum_{\boldsymbol{s} \in \mathcal{T}(\boldsymbol{x})} p_\theta(\boldsymbol{y} \mid \boldsymbol{s}) p_\phi(\boldsymbol{s} \mid \boldsymbol{x}) \right) \\ &= \log \mathbb{E}_{\boldsymbol{s} \sim p_\phi(\boldsymbol{s} \mid \boldsymbol{x})} \left[ p_\theta(\boldsymbol{y} \mid \boldsymbol{s}) \right] \\ &\geq \mathbb{E}_{\boldsymbol{s} \sim p_\phi(\boldsymbol{s} \mid \boldsymbol{x})} \left[ \log p_\theta(\boldsymbol{y} \mid \boldsymbol{s}) \right] \end{aligned}$$

## A.3 Experimental Setup and Hyperparameters

For all experiments the source parser is a neural PCFG [64] with 20 nonterminals and 20 preterminals. Other model settings that are shared across the experiments include: (1) Adam optimizer with learning rate $= 0.0005$, $\beta_1 = 0.75$, $\beta_2 = 0.999$, (2) gradient norm clipping at 3, (3) $L_2$ penalty (i.e. weight decay) of $10^{-5}$, (4) Xavier Glorot uniform initialization, and (5) training for 15 epochs with early stopping on the validation set (most models converged well before 15 epochs). The batch size is 4 for the SCAN and style transfer datasets, and 32 for the machine translation dataset. Due to memory constraints, in practice we use a batch size of 1 and simulate larger batch sizes through gradient accumulation. We observed training to be somewhat unstable and some datasets (e.g. SCAN and machine translation) needed training across 4 to 6 random seeds to perform well. In general we found it okay to overparameterize the grammar and use more nonterminals than seems necessary [13].

### A.3.1 SCAN

For SCAN, all models and embeddings are 256-dimensional. The QCFG has 10 nonterminals and 1 preterminal ($|\mathcal{N}| = 10, |\mathcal{P}| = 1$).Since SCAN does not provide official validations sets, we use the test set of the *simple* split as the validation set for the *add primitive (jump)*, *add template (around right)*, *length* splits. For the *simple* split, we use the training set itself as the validation set for early stopping. For decoding we take 10 sample derivations from the grammar and take the yield that has the lowest perplexity after rescoring.

### A.3.2 Style Transfer

On StylePTB, all models and embeddings with 512-dimensional. The QCFG has 8 nonterminals and 8 preterminals ($|\mathcal{N}| = 8, |\mathcal{P}| = 8$). We also contextualize the source embedding by passing it through a bidirectional LSTM, where the concatenation of the forward and backward hidden states for each word are projected down to 512 dimensions via an affine layer before they are fed to the TreeLSTM. For decoding we use 1000 samples and take the yield that has the lowest perplexity after rescoring. The adjective/verb emphasis tasks also provide a particular word in the source sentence to emphasize. We encode this information through a binary variable, whose embedding is added to the word embedding (before contexualization) in the encoder. Tokens which occur less than three times are replaced with the ⟨unk⟩ token.

The baseline uses a bidirectional LSTM encoder and an LSTM decoder with soft attention [8] and a pointer copy mechanism [95]. We tune over the number of layers, hidden units, and dropout rate. For decoding we use beam search with beam size of 5 (larger beam sizes did not improve performance).

### A.3.3 Machine Translation

In these experiments we use 512-dimensional models/embeddings. The QCFG has 14 nonterminals and 14 preterminals ($|\mathcal{N}| = 14, |\mathcal{P}| = 14$). As in the Style Transfer experiments, we also experiment with a variant where we contextualize the source word embeddings with a bidirectional LSTM before the TreeLSTM layer. For decoding we use 1000 samples and take the yield that has the lowest perplexity after rescoring. Tokens which occur less than two times are replaced with the ⟨unk⟩ token.

The LSTM baseline uses a bidirectional LSTM encoder and a LSTM decoder with soft attention [8]. We tune over the number of layers, dimensions, and dropout rate. The Transformer baseline is from OpenNMT [65], where we tune over the number of layers, hidden units, dropout rate, warm-up steps, and batch size. Given the small size of our dataset, the Transformer was particularly sensitive to the number of warm-up steps and the batch size. For decoding we use beam search with beam size of 5 (larger beam sizes did not improve performance).

We calculate BLEU with the multi-bleu.perl script from mosesdecoder. For the *daxy* test set, the original paper [68] only considers the "tu" translation of "you" to be correct. We follow Li et al. [70] and Chen et al. [19] and also count "vous" to be correct as well (e.g. both "tu n es pas daxiste ." and "vous n etes pas daxiste ." are considered to be valid translations of "you are not daxy .").