# OpenReview forum: "Sequence-to-Sequence Learning with Latent Neural Grammars"
_NeurIPS.cc/2021/Conference — NeurIPS 2021 Spotlight_

### Official Review · Reviewer_FyUp · 2021-07-15

**Rating:** 7
**Confidence:** 4

**Summary:**

This paper describes a sequence-to-sequence model that models both source and target trees as latent variables; the target tree is related to the source tree “quasi-synchronously.” The model is trained using REINFORCE.


**Limitations And Societal Impact:**

Given the complexity of the training method, I would like to see more discussion of training time and scalability.


**Main Review:**

# Model

The model is a pretty original departure from existing translation models, even those that use trees. I think it's great to see something like this working.

Given the complexity of the training method, I would really like to see some discussion of training time and scalability, like how many hours it takes to run the experiments in the papers, or what the largest parallel text you're able to train on is.

# Experiments

The model performs “respectably” on SCAN; compared with the other domain-agnostic methods, it’s clearly the best. What do the dashes in Table 1 mean? Is there any way to evaluate the methods above the line without SCAN-specific knowledge? Or to evaluate your method using comparable SCAN-specific knowledge?

The model also does well on several English rewriting tasks (e.g., active to passive). In Table 2 it might help to bold the best non-human score, which appears to almost always be your model. I think it would make sense to include a Transformer+copy baseline as well.

There is also an experiment on a *very* small machine translation task. Although the source for the data is cited, I think the size of the task should be stated more clearly: the sentences always begin with phrases like “I am,” “he is,” etc. and have at most 9 words each. Again, I think it would make sense to include a Transformer baseline here.

# Other

I feel that the introduction could be improved by being made more concise and getting more quickly to the really original part of the paper, the use of latent trees (currently not mentioned  until page 2). For example, “factorization of the output distribution via the chain rule coupled with a richly-parameterized neural network that models the local conditional distribution over the next element given the previous elements” seems to me like a very long way of saying “a neural model of p(y_t | y_1 \cdots y_{t-1})”.



**Time Spent Reviewing:**

1

---

> ### Author Response · Authors · 2021-08-08
> **Response**
>
> Thank you for your thoughtful review.
>
> - **Using SCAN domain-specific knowledge**: This is an interesting question/suggestion. We do think it may be possible to inject our method with domain-specific knowledge, for example we could manually add rules of the form “JUMP ⇒ jump” to force "JUMP" to always be transduced to "jump". This might result in much more data-efficient learning, since the model doesn’t have to learn to do this (the model does, however, learn to do this through training, as shown by the following table: https://imgur.com/a/r4yNboJ, which shows frequently-occurring rules in the training set). We will investigate this further.
>
>
> - **Transformer baseline**: Thank you for this suggestion. On the small MT dataset, the Transformer initially did quite badly due to optimization issues and hence we did not include this as a baseline in the initial submission. However, after some hyperparameter tuning (in particular, it was quite important to play around with the learning rate schedule and the batch size for this dataset), we were able to train a Transformer model that did quite well, and in fact outperformed our QCFG by 3~4 BLEU points. We will include this result and discuss it more in the next iteration of the paper.
>
> - **MT dataset**: We definitely agree that the MT dataset is *very* simple. However, the dataset does still contain some more “complex” sentences than just “i am [NP]/he is [NP]”, for example we have sentences such as:  “we are having trouble with our new neighbor”, “he is still grappling with religious beliefs”. On that note, when we tried to apply our approach to more realistic MT datasets, we quickly ran into memory issues due to the expensive dynamic program. Thus, while we believe the results are still interesting, scaling up our approach to practical settings is still very much a concern, as noted in the discussion section (though we find that this is generally the case for many neuro-symbolic approaches that perform well on SCAN). This is nonetheless an important limitation of the work and we will add more discussion around this point.
>
> - **Clarity/Formatting**: Thank you for your other suggestions regarding clarity/formatting. We will take note of this for the next iteration of the paper.

---

### Official Review · Reviewer_6KbN · 2021-07-17

**Rating:** 7
**Confidence:** 4

**Summary:**

The paper proposes a grammar-based method for sequence-to-sequence learning. The model is based on two latent grammars with neural parametrization. The source Probabilistic Context Free Grammar (PCFG) models the latent tree structure of the source sequence. The target Quasi (Probabilistic) Context Free Grammar jointly models the latent tree structure of the target sequence and the alignment between the source and the target tree. Both source and target parses need to be marginalized out in order to train the model. The proposed method performs approximate marginalization by sampling the source trees and then exactly marginalizing the target trees with an algorithm of a high yet polynomial computational complexity. The algorithm achieves high performance on the compositional tasks from SCAN, style-transfer and small-scale translation tasks.

**Limitations And Societal Impact:**

The paper is quite up-front about the limitations of the proposed method, such as conditional independence assumptions and computational complexity.


**Main Review:**

Quality: I find the presented results sufficient for publication. The quantitative results are better than adequately chosen baselines. The qualitative results give useful insights into the inner workings of the method.

Further quality improvements can be achieved:
- More synthetic compositional generalization datasets could be used (COGS, CFQ, CLOSURE)
It would be nice to see semantic parsing experiments using natural data. In particular, a lot of similar methods (e.g. [46])  have recently been tested on GeoQuery.
- Some ablation results would be nice to have. What if a linear source parse is used?
- Error analysis for SCAN

Clarity: the paper is well-written and clear for people with a bit of background in grammars and computational linguistics. To make it more accessible for a wider audience, I would recommend including pseudocodes for the target parse marginalization and for the inference.

Originality: there is a fair amount of work on more compositional seq2seq transduction algorithms, some of which is using grammars (e.g. [46]). The paper appears sufficiently distinct for other works, but a deeper discussion on its similarities/differences with other grammar-based approaches would not hurt.

Significance: the fact that a grammar-based model with so many latent variables can be trained and can perform well is a significant result. More experiments could further increase significance.


**Time Spent Reviewing:**

3

---

> ### Author Response · Authors · 2021-08-08
> **Response**
>
> Thank you for your thoughtful review.
>
> - **Semantic Parsing datasets**: This is a great suggestion. We did not apply our approach to semantic parsing datasets because our use of unary/binary rules only makes it somewhat unnatural to capture cases where bracketing information is explicitly written down on the target side, which is the case for many existing semantic parsing datasets. For example, consider the following target example from COGS: “* captain ( x _ 1 ) ; eat . agent ( x _ 2 , x _ 1 )”. While it is theoretically possible to capture the underlying structure with a binary tree, it is quite awkward. For example, “( x _ 1 )” would need to be bracketed by the QCFG as something like “{ ( { { x _ } 1 } ) }”, where we have used curly brackets {/} for the QCFG structure and regular brackets (/) for the COGS structure that is explicitly given. This could be overcome by treating phrases such as “x _ 1” as an entire token, but this would require some dataset specific processing. Nonetheless we agree that it would be interesting to adapt our approach to semantic parsing.
>
> - **Ablation**: Based on the suggestion we tried two additional ablations on the MT dataset : (1) Using a linear source parse (i.e. either left/right branching trees): this results in ~3 BLEU decrease, and (2) Using no constraints on the rule set: this results in ~1.5 BLEU point decrease. This indicates that both the use of learned source trees and the addition of inductive bias via constraints on the rule set are important elements of our approach.
>
> - **Error analysis on SCAN**: We found that the model often made errors in the initial root node, where it failed to make sure that “A and B” are split into “A” and “B”. See an example here: https://imgur.com/a/DChNSuf. Manually fixing this during decoding improves performance by ~2%, and highlights an additional advantage of our approach---since we have more control into the generation process via the nonterminals, we can fix systematic errors by manipulating the nonterminals during decoding. We also did an additional analysis of the learned rules for SCAN, and have posted them here: https://imgur.com/a/r4yNboJ
>
> - **Clarity**: Thank you for your suggestion. We will take note of this.

---

> > ### Comment · Reviewer_6KbN · 2021-08-11
> > **thanks**
> >
> > Thanks for your response. The new ablation results are satisfactory. I encourage you to add the discussion about using your method for semantic parsing to the final version of the paper.

---

### Official Review · Reviewer_fSHR · 2021-07-18

**Rating:** 7
**Confidence:** 5

**Summary:**

This paper investigates an neural approach of probabilistic quasi-synchronous context-free grammars (QCFG) and its application in various sequence-to-sequence applications.  QCFG is a type of very general tree-to-tree synchronous grammar where any subset of nodes in the source tree can be aligned to a node in the target tree.  This paper proposes an neural approach to QCFG, where the representation of tree nodes are parameterized with a TreeLSTM model. Further, the paper conducts a joint training of a source parser and the QCFG, where the source syntax tree is a binary tree without labels in the nodes.  A complex combination of techniques including Monte Carlo estimation, dynamic programming, MAP infrerence, approximate decoding is used to deal with the complexity in training and inference.  Further extensions are proplsed to deal with the phrase-to-phrase copying, domain specific constraints problems and adding autogressive language models.  Experimewnts are conducted in three tasks: natural language to commands, style transfer, and machine translation.  All experiments show significant improvement over previous work, especially seq2seq-LSTM models.

**Main Review:**

Pro:

A novel seq2seq neural model with latent syntax models.  The model can be trained solely in parallel texts without syntax treebanks.  Experiments show improvements on various tasks.
Case studies demonstrate that the proposed method better model the structural alignents between the source and target texts and have better performance in compositional problems.

Cons:

The experiments are all in quite small scale.  I wonder if it works the same well with large training data.
The computational cost is rather high.  I hope to see the training time of these models.


Suggestions:

Line 148-153 Incorporating autogressive language models: not very clear to me.  More explanation or an example is needed.
Section 3.3: The seq2seq-LSTM is a bit weak as a baseline.  Transformer should be added.


**Time Spent Reviewing:**

6

---

> ### Author Response · Authors · 2021-08-08
> **Response**
>
> Thank you for your thoughtful review.
>
> - **Small scale data + computational cost**: We definitely agree that our experiments are on very small datasets. When we tried to apply our approach to more realistic MT datasets, we quickly ran into memory issues due to the expensive dynamic program. Thus, while we believe the results are still interesting, scaling up our approach to practical settings is still very much a concern, as noted in the discussion section (though we find that this is generally the case for many neuro-symbolic approaches that perform well on SCAN). This is nonetheless an important limitation of the work and we will add more discussion around this point.
>
> - **Clarity regarding autoregressive models**: We agree this section should be explained better. We will update the paper to make the section clearer, with concrete examples for why incorporating an autoregressive model might be useful.
>
> - **Transformer baseline**: Thank you for this suggestion. On the small MT dataset, the Transformer initially did quite badly due to optimization issues and hence we did not include this as a baseline in the initial submission. However, after some hyperparameter tuning (in particular, it was quite important to play around with the learning rate schedule and the batch size for this dataset), we were able to train a Transformer model that did quite well, and in fact outperformed our QCFG by 3~4 BLEU points. We will include this result and discuss it more in the next iteration of the paper.

---

> > ### Comment · Reviewer_fSHR · 2021-09-10
> > **The response is fine with me.  I will raise my rating to 7.**
> >
> > The response is fine with me.  I will raise my rating to 7.

---

### Official Review · Reviewer_e5Ff · 2021-07-20

**Rating:** 9
**Confidence:** 4

**Summary:**

This paper proposes a framework for sequence-to-sequence learning with synchronous grammars. In this framework, both the source trees and target trees are modeled as latent variables, and each node in the target tree is transduced by a subset of nodes in the source tree. These latent trees are marginalized out during training and no extra supervision is required. The grammars are parameterized by neural parameters so that the model can be trained end-to-end without manual feature engineering. Such hierarchical, latent seq2seq modeling makes the generation process more interpretable than typical seq2seq models like LSTM or transformers, as well as providing an interface to add inductive bias easily, for which the paper has included several examples. The experiments are focused on settings where generalization is difficult (e.g. compositional generation or when the training data is small),  on language navigation, style transfer, and machine translation tasks the proposed model outperforms traditional seq2seq models substantially, demonstrating better sample efficiency.


**Ethics Review Area:**

["I don’t know"]

**Limitations And Societal Impact:**

Yes

**Main Review:**

I’ll detail the strengths, weaknesses, and questions below:
### Strengths:

1. I think the direction is very interesting and important. Currently, most neural models are data-driven without explicit inductive bias, which is uninterpretable and often with low sample efficiency. Therefore, in many domains/tasks generalization (e.g. compositional generalization) is of great interest to the community. While there are some methods like data augmentation to improve generalization, this paper proposes a more principled method and tackles the issue from the model framework itself. Therefore, I think this paper stands as a strong contribution to propose a new model class and show its effectiveness, and could potentially open a new path for future work to study generalization from a different perspective.


2. The techniques are novel in modern seq2seq tasks. While the concept of QCFG is not new, the neural parameterization, implementation on recent seq2seq benchmarks, and strong results compared to traditional neural seq2seq models are non-trivial efforts.


3. In Section 2.4, the examples on how to imbue inductive bias and connection to some common techniques in NN are very intriguing, there could be future interesting work on this aspect to add different inductive bias on various tasks/datasets.


4. Empirical results are strong.


5. The paper is very well-written, I enjoyed reading it a lot.


### Weaknesses:

The main weaknesses are the complexity of the method and high time complexity due to the dynamic program, which could limit the method in large-scale settings as acknowledged by the authors.


**Time Spent Reviewing:**

2

---

> ### Author Response · Authors · 2021-08-08
> **Response**
>
> Thank you for thoughtful review.
>
> We definitely agree that computational complexity of the approach is a serious limitation of the proposed approach. Scaling the QCFG realistic seq2seq dataset will likely require significant modifications. One direction might be to explore a "soft" version of the grammar, where the nonterminals embeddings are contexualized against source elements via soft attention. The difference between the two would be analogous to the difference between soft/hard attention in standard seq2seq models. In this case, the runtime and memory for marginalizing over target trees in this soft QCFG would no longer depend on source length. Yet another direction could involve using something like state dropout (as in https://arxiv.org/pdf/2011.04640.pdf). We will add more discussion around this point in the next iteration of the paper.

---

### Decision · Program_Chairs · 2021-09-28

**Decision:**

Accept (Spotlight)

**Comment:**

The paper proposes a neural approach of probabilistic quasi-synchronous context-free grammars (QCFG) and evaluates its application to various sequence-to-sequence applications (natural language to command, style transfer, and machine translation). While tree-based, the approach doesn’t require any treebank or other linguistic annotation, and training is end-to-end without any manual feature engineering.

Overall, reviews find the paper to be quite strong. While there is prior work on quasi-synchronous grammars, the neural parameterization of the paper, the non-reliance on feature engineering, the application to recent benchmarks, and strong results are seen as significant contributions. The overall direction of this work is important compared to current fully data-driven models, which typically do not incorporate any inductive bias and can cause low sample efficiency and over-reliance on surface-form information.

The only main weaknesses of the paper are high time complexity and experiments that are overall small scale. But I think it is reasonable to downplay these limitations, as one of the stated goals of the paper is sample efficiency rather than ability to efficiently train on large datasets.


**Consistency Experiment:**

NeurIPS has a long history of experimentation. In 2014, NeurIPS ran an experiment in which 10% of submissions were reviewed by two independent committees to quantify the randomness in the review process. This year, we repeated a variant of this experiment to see how the quality of the review process has changed over time.  This paper was part of the experiment and was therefore assigned to two committees (consisting of reviewers, an Area Chair, and a Senior Area Chair) that reached independent decisions.  If both committees made the same recommendation, this recommendation was followed. If a single committee recommended acceptance, the paper was accepted (with the exception of a few cases in which the other committee identified what we considered a fatal flaw, e.g., an error in a key result).

This copy’s committee reached the following decision: **Accept (Spotlight)**

The other committee assigned to the paper recommended **Accept (Poster)**.  You can find the other set of reviews, along with any follow up discussion with the authors here:
https://openreview.net/forum?id=pbfAgoc_l2w